# Finite Element Analysis of Novel Separable Fixture for Easy Retrievement in Case with Peri-Implantitis

**DOI:** 10.3390/ma12020235

**Published:** 2019-01-11

**Authors:** Won Hyeon Kim, Eun Sung Song, Kyung Won Ju, Jong-Ho Lee, Man Yong Kim, Dohyung Lim, Bongju Kim

**Affiliations:** 1Clinical Translational Research Center for Dental Science, Seoul National University Dental Hospital, Seoul 03080, Korea; wonhyun79@gmail.com (W.H.K.); songeunsung@gmail.com (E.S.S.); dentykim@gmail.com (M.Y.K.); 2Department of Mechanical Engineering, Sejong University, Seoul 05006, Korea; 3Dental Research Institute, Seoul National University, Seoul 03080, Korea; kwcindyju@hotmail.com (K.W.J.); leejongh@snu.ac.kr (J.-H.L.); 4Department of Oral and Maxillofacial Surgery, School of Dentistry, Seoul National University, Seoul 03080, Korea

**Keywords:** peri-implantitis, dental implants, biomechanics, finite element analysis, separable fixture

## Abstract

Peri-implantitis is a common complication following dental implant placement, which may lead to bone loss and fixation failure. With the conventional fixture, it is difficult to perfectly clear-up the infection. To solve this, we have designed a separable fixture of which the top part is replaceable. This study aimed to compare the structural and biomechanical stability of the separable and conventional fixture. A single surgical model corresponding to the first molar in a virtual mandible model and conventional/separable implants were reproduced to compare the biomechanical characteristics of the implants using finite element analysis (FEA). The loading condition was 200N preload in the first step, and 100N (Axial), 100N (15°), and 30N (45°) in the second step. The stress distribution on the cortical bone in the separable implant was lower than the conventional implant. In particular, the Peak von Mises Stress (PVMS) values of the separable implant under lateral load was found to be about twice as low as that of the conventional implant. In this study, we suggest that the separable implant has an equivalent biomechanical stability compared to the conventional implant, is easy to retrieve in the case of peri-implantitis, and has an excellent initial stability after the surgery when used in stage 2.

## 1. Introduction

Over the past several decades, treatment with dental implants has been widely used to restore oral masticatory movement when natural teeth are lost or partially damaged [1,2,3]. Previous studies showed that the success rate was 95% or more when properly designed and manufactured dental implants were placed on the defect site as intended [4,5], and 85~95% with partial dentures (bridge) [6,7]. The majority of dental implants have a high success rate in the long-term [8,9,10]. Nonetheless, problems such as postoperative complications, union, and fixation failures occur. There are various factors to implant fixation failure, such as screw loosening between the fixture and abutment, prosthesis breakage, and peri-implantitis [11,12,13]. In particular, peri-implantitis, which is one of the major causes, occurs due to inflammation in the hard and soft tissues around the implant and leads to hemorrhage and bone loss [13,14]. To prevent and treat peri-implantitis, various studies are currently being developed in the dental field, such as local or systemic antibiotics injection, surgical treatments, and an implant surface of titanium with antimicrobial properties [15,16,17,18]. Treatment methods such as local/systematic antibiotics, laser therapy, ultrasonography, and metallic curettes were effective in the removal of a bacteria biofilm on the implant surface, but not in cases of severe infection due to difficulties in accessibility and visibility [19,20]. As peri-implantitis is caused by the accumulation of dental plaque and bacteria biofilm formation leading to alveolar bone loss on the gingival implant surface [21], it is difficult for the patient to notice its occurrence. Also, Heasman et al. reported that among implants placed for 9~11 years, the cause of peri-implantitis was 28~56% due to the patients and 12~43% due to the implants [22]. For these reasons, patients with dental implants regularly need to check for inflammation on the implant surface and receive treatment, but there is still a lack of any clear treatment methods to prevent and treat peri-implantitis. 

The conventional implant fixture which is commonly used consists of one body where contamination on the surface is removed and guided bone regeneration (GBR) is performed in the case of peri-implantitis [20], but there is a lack of evidence on the effectiveness of this method [23]. Moreover, there is a high risk of GBR failure since it is difficult to completely clear-up the contaminated area of the fixture. Considering these risks, we have designed a fixture that is separable in the top and bottom parts, so that, in the case of peri-implantitis, the whole contaminated top part of the implant could be replaced. As the separable implant newly replaces the whole contaminated part of the fixture, it would not only increase the success rate of GBR, but would also be convenient and reduce time.

In a study on marginal bone loss of 131 implant prostheses by Cappiello et al., 1.67 ± 0.37 mm marginal bone loss occurred for platform matching implants and 0.95 ± 0.32 mm for platform switching implants [24]. Likewise, marginal bone loss after an implant is inevitable, and aesthetic problems may arise if it occurs on the anterior.

The conventional fixture is a one-body structure, so it is difficult to remove the implanted fixture if peri-implantitis occurs. The contaminated top surface of the implant is treated using the methods mentioned above, but it is also difficult to perfectly clear-up the infected area. Therefore, we have designed a separable fixture which divides the body into two parts (top and bottom) so that only the infected top part has to be replaced if the surface is contaminated.

As the separable fixture consists of top and bottom parts compared to the conventional fixture, it may have the disadvantage of a low stability due to the small gap at the junction between the top-bottom parts when attached to the abutment, crown, and abutment screw. Yet, there is no reference for comparison to the conventional fixture since no cases have been reported so far with such a fixture. Therefore, an analysis to compare the structural stability of the conventional and separable fixture is required. Previous studies have performed comparative biomechanical evaluations using FEA to analyze various changes, such as in design and load [25,26,27,28,29,30,31].

This study was conducted to compare the biomechanical stability of the easily separable fixture in the case of peri-implantitis and the commonly used conventional fixture using FEA. The hypotheses of this study are (1) the biomechanical stability of the separable and conventional fixture is equivalent; and (2) compared to the conventional fixture, the separable fixture consists of top and bottom parts, and the load distribution and structural stability are superior when a lateral load is applied on the tooth after osseointegration. 

## 2. Materials and Methods

### 2.1. Designs and Dimensions of Conventional and Separable Fixtures

The exterior designs of the conventional and separable fixtures were identical, and were 6 mm in length, 3.4 mm in diameter, and 0.4 mm screw thread pitch (Figure 1a,b). For the abutment screw, the head diameter was 2.5 mm, body diameter 1.6 mm, length 7.5 mm, and 0.35 mm screw thread pitch (Figure 1c). The abutment of the same design was applied for conventional and separable fixture models (Figure 1d). The implant model of this size was selected as it is the worst case with the smallest fixture.

### 2.2. Finite Element Analysis (FEA)

The implant systems, including the abutment, fixture, and abutment screw of two different fixtures, were designed as a three-dimensional CAD model using SolidWorks software (Solidworks 2016, Dassault Systèmes SolidWorks Corp., Waltham, MA, USA) (Figure 1). The crown model was constructed by extracting the tooth shape corresponding to the first molar from the cone beam computed tomography (CBCT) images. 

To compare the stress distribution and biomechanical stability within the bone between the conventional and separable fixtures, a CBCT image of a normal Korean male adult skull was cut into a 0.25 mm thickness to obtain two-dimensional images, and the obtained images were used to reconstruct three-dimensional mandible models using the Mimics program (Mimics Research v19.0, materialise, Leuven, Provincie Vlaams-Brabant, Belgium) (Figure 2a,b). As previous literature reported that masticatory force is mainly loaded in the molars during mastication, this study constructed a single tooth model by extracting the portion corresponding to the first molar [32]. The shape of the bone was reproduced as distinguished cortical and cancellous bones, and the cortical bone was constructed to have a uniform thickness of 2 mm with reference to previous literature (Figure 2c) [33]. An FEA program (ABAQUS CAE2016, Dassault systems, Vélizy-Villacoublay, Yvelines, France) was used to construct the single tooth model into surgical models for both conventional and separable fixtures (Figure 2c), and the Hypermesh program (Altair Hyperworks v17.0, Altair Engineering Inc., Troy, MI, USA) was used to form a mesh of the models. The properties of the cortical and cancellous bones, crown, and implant were applied by referring to previous literature (Table 1) [25,26,27,34].

A study by Joao Paulo performed FEA with a short dental implant and derived an optimum mesh size of 0.3 mm through mesh convergence [35]. Accordingly, the maximum mesh size of the implant (abutment, fixture, abutment screw) in this study was set at 0.15 mm, which is lower than the previous study, and the contact interface between the bone and the implant was 0.15 mm. The numbers of elements, nodes, and mesh sizes used in this study are shown in Table 2.

The numbers of elements and nodes for the finite element surgical model are shown in Table 2. For the boundary conditions, the surgical model was applied to the complete restraint condition at 6 degrees of freedom (DOF) so that both sides and distal parts of the cortical and cancellous bone were not rotated and moved in any direction (Figure 2c). The interface of the cortical and cancellous bone, bones and fixture, and abutment and crown were applied as “Tie contact”. The “Tie contact” type simulated perfect osseointegration in which the implant and the surrounding bone were fully integrated so that neither sliding nor separation in the implant-bone interface was possible [27]. The interfaces between the fixture, abutment, and abutment screw were applied as “Frictional contact”. A friction coefficient of 0.5 was used [27]. Frictional contact implies that a gap between the implant parts might exist under an occlusal force [27].

A preload of 200N was applied vertically to the abutment screw to assume the condition of the fixation with the abutment and fixture [27]. Then, considering the load generated at various angles in the oral cavity, vertically 100N [28], 100N at 15° [29], and 30N at 45° [28] of load were applied at the first molar (Figure 3). The applied load was within the physiological range [36], and various loading directions were applied as non-axial loading affects major remodeling at the interface between the bone and implant [37].

The maximum equivalent stress (Max EQS) at the bone surface around the implants was derived [38]. A lower Max EQS means that the stress distribution on the bone surrounding the fixture would be lower, the failure risk of bone structure would be less, and the success rate would be higher [38]. Finally, the failure risk of implants (abutment, fixture, abutment screw) was analyzed by comparing the PVMS in the implants calculated by FE analysis to the yield strength (483 MPa) of the titanium grade 4 and the yield strength (793 MPa) of the titanium grade 5 [39]. 

### 2.3. Statistical Analysis

The Max EQS values of the cortical and cancellous bone of each node at the contact surface between the bone and fixture were presented as mean ± standard error of the mean (SEM), and two independent sample two tailed t-test was performed using the means for comparison between conventional and separable fixtures. The level of significance was set at *p*-value < 0.05. Statistical analysis was performed using SiamaPlot (Systat Software Inc., San Jose, CA, USA).

## 3. Results

The mean Max EQS values of both the cortical and cancellous bone surrounding the fixture after vertical loading and oblique loading are shown in Figure 4a,b. 

In cortical bone, the stress distribution was higher in the conventional fixture than the separable fixture under a vertical loading condition (*p* < 0.001). However, there was no significant difference between conventional and separable fixtures in both the 15° and 45° loading condition, respectively (*p* = 0.368, *p* = 0.535) (Table 3). In cancellous bone, it was identified that the results of stress distribution were statistically significant for all loading directions. The results of the separable fixture were higher than the conventional fixture under a vertical loading condition (*p* < 0.01). Similar to the vertical load, the results showed that the values of the separable fixture were higher than the conventional fixture under oblique loading conditions (15° and 45°, respectively) (*p* < 0.05, *p* < 0.01) (Table 3).

In order to compare the difference between the conventional and the separable fixture, the stress distribution at the region where the fixture and the abutment were contacted was compared under the various loading conditions. The mean Max EQS, on both the abutment outer surface and fixture inner surface at the junction part, after vertical loading and oblique loading, is shown in Figure 4c,d.

In the abutment outer surface, the conventional fixture was significantly greater than the separable fixture in both 15° and 45° loading, respectively (*p* < 0.001). However, there was no significant difference between the conventional and separable fixtures in vertical loading (*p* = 0.108) (Table 3). In the fixture inner surface, the stress distribution of the conventional fixture was significantly greater than the separable fixture under all loading conditions (*p* < 0.001) (Table 3).

The PVMS values of the implants (abutment, fixture, abutment screw) are summarized in Table 4. The PVMS values of the abutment and abutment screw were higher in the separable fixture than the conventional fixture under the vertical loading condition. However, the PVMS value of the fixture in the conventional fixture was shown to be higher than the separable fixture. On the other hand, the PVMS value under the oblique loading condition showed a different trend of PVMS values from the vertical load. The stresses of abutment and abutment screw in oblique loading were higher in the conventional fixture than the separable fixture. Especially, the stresses of the abutment were identified as about two times greater in the conventional fixture than the separable fixture.

## 4. Discussion

When peri-implantitis occurs after dental implant placement, treatments such as antibiotics injection and metallic curettes are used to remove contamination on the surface of the conventional fixture, but it is difficult to clear-up perfectly [19,20]. For this reason, this study designed a fixture that is separable in top and bottom parts so that in the case of peri-implantitis, the contaminated top part could be removed and replaced with a new one. As only the contaminated part is replaced, it is surgically more convenient compared to the conventional fixture. Also, it has a higher success rate of clearing up compared to the conventional method, hence it is considered to provide a higher success rate of peri-implantitis treatment.

This study compared stress on the bone surrounding the fixture under various loading conditions to assess the biomechanical stability and compared PVMS of the implants to predict the tendency of implant failure using FEA. For decades, FEA has been used to suggest solutions for problems of a complex geometrical shape in industries [40]. FEA in implant dentistry was first used in 1976 for stress analysis between the bone and implant; subsequently, it has been rapidly applied for biomechanical evaluation [30]. In this study, two loading steps were used [27]. The first loading step applied the preload for complete tightening of the abutment screw. Then, occlusal force was applied at the crown. High stress was generated on the third screw thread position of the abutment screw in both conventional and separable surgical models when only the preload was applied (Figure 5e). In particular, the conventional fixture showed a tendency of stress concentration at the third thread position, while the separable fixture showed a smooth stress distribution compared to the conventional fixture. In this way, excess stress applied on the abutment screw in the conventional fixture was considered to result in fatigue fracture of the abutment screw if the load was applied at various directions for a long period of time.

The mean Max EQS value on the cortical bone surrounding the fixture was statistically lower in the separable fixture than the conventional fixture at vertical load, but there was no statistical difference between the conventional and separable fixtures at oblique load. A lower Max EQS of the separable fixture in a vertical loading condition indicated less stress on the surrounding bone, which implied less damage to the cortical bone and, subsequently, a higher implant success rate [38]. Both conventional and separable fixtures showed a tendency of stress concentration at the cortical bone, and this is because the elasticity modulus of the cortical bone is 10 times or more than that of cancellous bone (Figure 5a,b). On the other hand, the mean Max EQS value of the cancellous bone was statistically higher in the separable fixture than the conventional fixture in all loading directions. For this reason, stress was transferred more desirably to the cancellous bone in the separable fixture than the conventional implant, and it may reduce the stress concentration in the cortical bone.

To compare stress distribution on the abutment-fixture connection surfaces of two different fixture designs, the mean Max EQS at the contact area was derived. At vertical load, the mean stress at the outer surface of the abutment did not show any statistical difference, but it was higher in the conventional fixture than the separable fixture at oblique stress. The mean stress on the inner surface of the fixture was higher in the conventional fixture than the separable fixture in all loading conditions. Moreover, to compare the biomechanical stability of the implant, the PVMS of each abutment, fixture, and abutment screw were derived. At a vertical loading condition, the conventional fixture showed less stress than the separable fixture on the abutment, but more on the fixture. On the other hand, the oblique load showed a different tendency compared to the vertical loading condition. The conventional fixture showed higher stress on the abutment than the separable fixture, and lower stress on the fixture. There was not much difference in stress between the conventional and separable fixtures at vertical load, but it was approximately two times higher on the abutment in the conventional fixture than the separable fixture. In particular, excess stress was concentrated on the hexagonal structure where tension is generated while it resists lateral force on the contact surface between the conventional fixture and the abutment (Figure 5c). On the contrary, stress was better transferred between the fixture and the abutment in the separable implant. However, the maximum stress values of all implants were lower than the yield stress values of each material, which were 483 MPa (Ti grade 4) and 793 MPa (Ti grade 5) [39], so the risk of fracture is expected to be low. On the other hand, the load was applied diagonally or sideways at oblique load, so stress of the fixture was highest at the region of bending load between the fixture and the abutment (Figure 5). Yang [31] reported that implant structure and loading conditions affect the stress distribution on the tissues around the implant, and it was also confirmed in this study that the pattern of stress distribution differs according to the loading conditions. As the top structure of the separable implant allows micro-movement, it also showed a tendency to bend as the abutment bens if oblique force was applied. This was found to create a greater contact surface area between the abutment and the fixture. Because the contact area is larger in the separable implant than the conventional one at lateral load, it is considered that stress concentration on the abutment would be reduced and biomechanical stability would be better after surgery. 

A typical dental implant procedure consists of two stages (stage 1 and 2). In methods used during stage 1, where premature loading is applied, micro-movement during initial implant placement may disrupt the formation of new tissue and form fibroblasts rather than osteoblast mineralization [36]. For this reason, the separable implant is considered to have the disadvantage of less initial stability. However, if the separable fixture was used in stage 2, prosthesis would be conducted after complete osseointegration through the healing cap, so the postoperative biomechanical and structural stability of the implant would be expected to be higher compared to the conventional implant. 

As Cappiello et al., reported [24], marginal bone loss is inevitable after implant placement, so a bone graft is required for conventional fixtures, whereas the separable implant is considered more advantageous for its convenient treatment. The top part could be removed and a hexagonal, external type abutment could be attached to the bottom part, or a top part of a shorter length could be replaced.

In this study, FEA was performed to compare the biomechanical performance of the conventional and separable fixture models using a single surgical model under static loading. However, since the complex and fatigue loading on the tooth is important, it is necessary to analyze the characteristics and fracture pattern of the fatigue of the conventional and separable fixture models within a clinical fatigue loading condition in the actual implant system. 

## 5. Conclusions

On the basis of the results, cortical bone damage is considered to be lower in the separable fixture than the conventional at vertical load, and the separable fixture system is considered to be superior to the conventional system at oblique load, because there was no significant difference between the conventional and separable fixtures. However, micro-movement of the separable fixture could result in fibroblast formation instead of ossification in implant stage 1, so it is recommended for use in stage 2. Moreover, like the hypotheses of this study, the separable fixture was more stable in terms of the implant structure at lateral load than the conventional fixture. Through this study, the equivalence of the separable fixture system and the conventional system was demonstrated, but further pre-clinical/clinical research is required for data such as on osteogenesis of the conventional and separable fixtures. 

## Figures and Tables

**Figure 1 materials-12-00235-f001:**
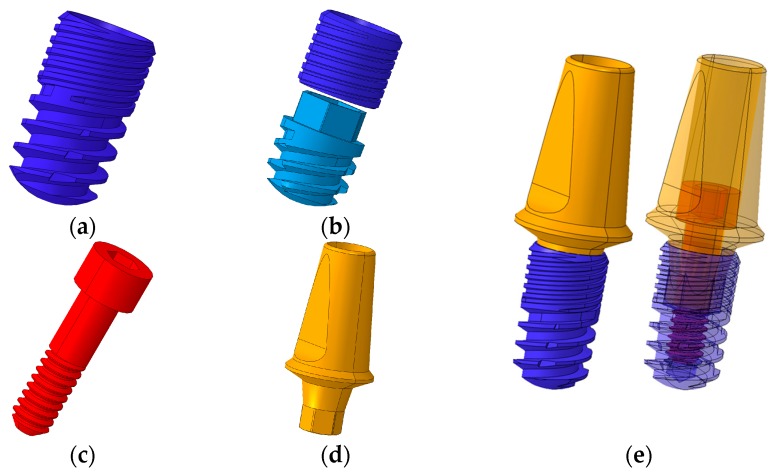
3D CAD model of dental implant system. (**a**) Conventional fixture type, (**b**) separable fixture type, (**c**) abutment screw, (**d**) abutment, and (**e**) assembled model of implant system (Left: exterior design of the fixture/Right: internal structure of the fixture).

**Figure 2 materials-12-00235-f002:**
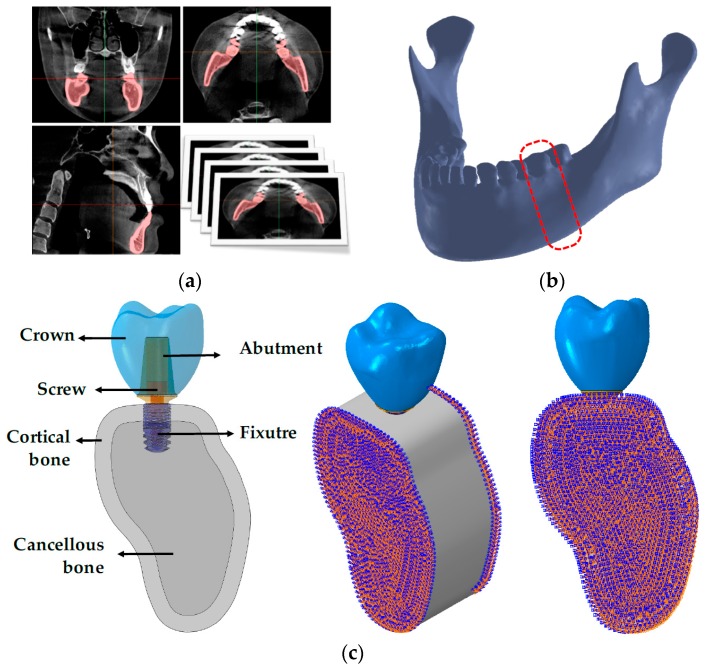
Modeling of three-dimensional finite element for single surgical model. (**a**) Geometry data extraction from CBCT images, (**b**) reconstruction of surface and volume for mandible bone, and (**c**) finite element single surgical model for conventional and separable fixtures.

**Figure 3 materials-12-00235-f003:**
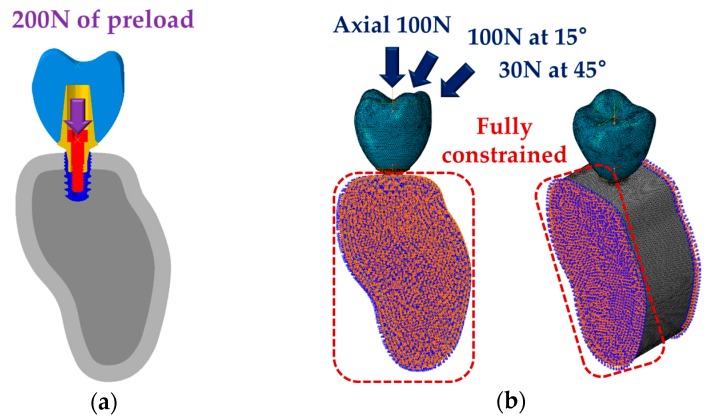
Loading and boundary conditions of single surgical models (conventional and separable fixture types). (**a**) Application of preload of 200N at abutment screw assuming the connection state of abutment and fixture. (**b**) Application of 100N (axial), 100N (15°), and 30N (45°) considering the various directional loads applied to teeth.

**Figure 4 materials-12-00235-f004:**
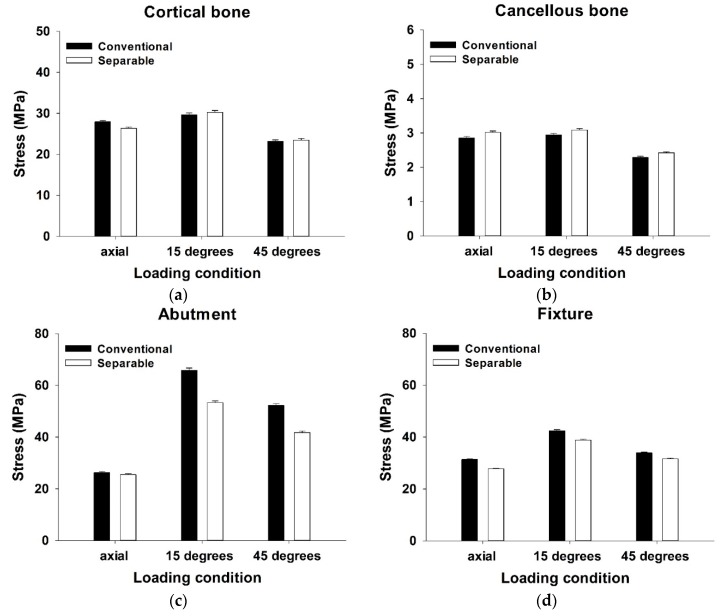
Maximum equivalent stress (Max EQS) in (**a**) cortical bone, (**b**) cancellous bone, (**c**) abutment outer surface, and (**d**) fixture inner surface under loading of 100N (axial), 100N (15°), and 30N (45°).

**Figure 5 materials-12-00235-f005:**
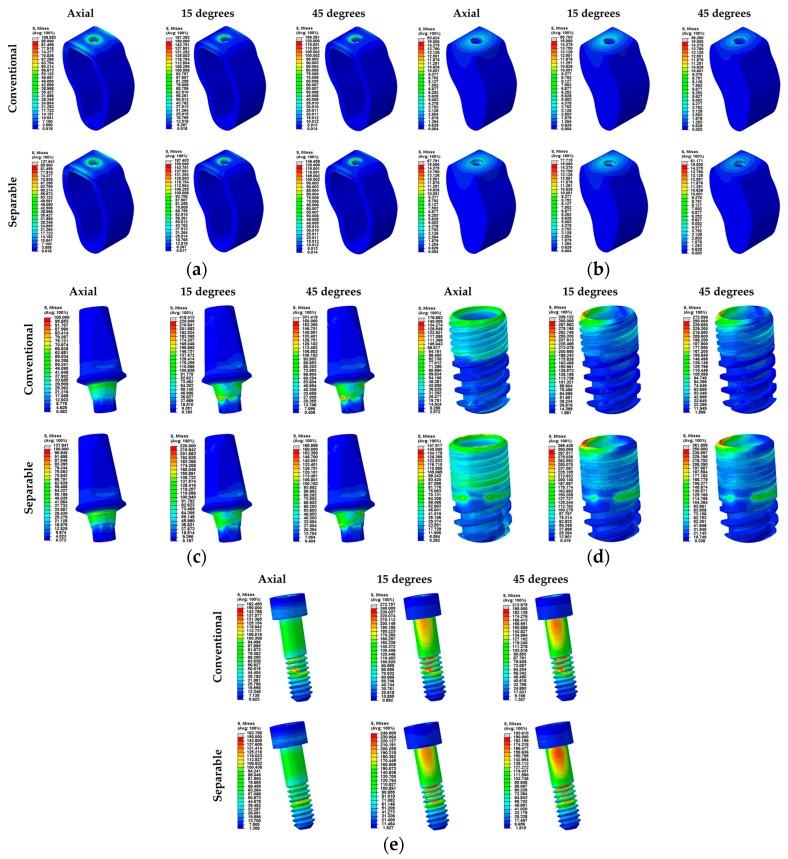
Stress distribution of each part under various loads (Axial, 15°, and 45°). (**a**) Cortical bone, (**b**) cancellous bone, (**c**) abutment, (**d**) fixture, and (**e**) abutment screw for conventional and separable fixtures.

**Table 1 materials-12-00235-t001:** Material properties of finite element models.

Components	Young’s Modulus (MPa)	Poisson’s Ratio
Crown (Zirconia) [34]	205,000	0.19
Abutment (Ti-grade 5) [26]	114,000	0.33
Fixture (Ti-grade 4) [25]	105,000	0.34
Abutment screw (Ti-grade 5) [26]	114,000	0.33
Cortical bone [27]	13,000	0.30
Cancellous bone [27]	690	0.30

**Table 2 materials-12-00235-t002:** Number of elements and nodes for finite element surgical model.

	Elements		Nodes		Mesh Size (mm)	
	Conventional Fixture	Separable Fixture	Conventional Fixture	Separable Fixture	Maximum	Minimum
Crown	121,740		25,678		0.30	0.15
Abutment	88,637		21,139		0.15	0.05
Fixture	125,966	48,692 (up) 104,432 (down)	24,396	12,300 (up) 23,594 (down)	0.15	0.05
Abutment screw	60,588		13,836		0.15	0.03
Cortical bone	287,389	295,533	62,917	64,492	1.00	0.15
Cancellous bone	275,822	273,597	57,108	56,765	1.00	0.15

**Table 3 materials-12-00235-t003:** FEA results (Mean ± SEM) of the conventional and separable fixtures for cortical bone, cancellous bone, abutment, and fixture (* *p*-value < 0.05, ** *p*-value < 0.01).

	Direction	Maximum Equivalent Stress (MPa)				
		Conventional Fixture	Number of Node	Separable Fixture	Number of Node	*p*-Value
Cortical bone	100N (Axial)	27.97 ± 0.28	3881	26.38 ± 0.26	3993	<0.001 **
	100N (15°)	28.88 ± 0.46		29.73 ± 0.47		0.368
	100N (45°)	22.89 ± 0.37		23.32 ± 0.37		0.535
Cancellous bone	100N (Axial)	2.85 ± 0.05	4745	3.02 ± 0.04	5696	0.008 **
	100N (15°)	2.94 ± 0.05		3.09 ± 0.04		0.019 *
	100N (45°)	2.29 ± 0.04		2.42 ± 0.03		0.006 **
Abutment outer surface	100N (Axial)	26.32 ± 0.33	3569	25.49 ± 0.40	3569	0.108
	100N (15°)	65.81 ± 0.91		53.29 ± 0.67		<0.001 **
	100N (45°)	52.23 ± 0.72		41.74 ± 0.53		<0.001 **
Fixture inner surface	100N (Axial)	31.34 ± 0.18	7109	27.79 ± 0.18	7311	<0.001 **
	100N (15°)	42.46 ± 0.36		38.84 ± 0.32		<0.001 **
	100N (45°)	33.83 ± 0.29		31.57 ± 0.26		<0.001 **

**Table 4 materials-12-00235-t004:** PVMS results of the conventional and separable fixtures for abutment, fixture, and abutment screw.

	PVMS (MPa)					
	Abutment		Fixture		Abutment Screw	
	Conventional Fixture	Separable Fixture	Conventional Fixture	Separable Fixture	Conventional Fixture	Separable Fixture
100N (Axial)	95.853	137.64	176.86	147.52	162.45	163.79
100N (15°)	418.41	218.87	339.13	395.43	272.76	239.09
100N (45°)	339.13	165.99	272.59	361.86	212.98	193.61

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
