# Peer review of "Finite Element Analysis of Novel Separable Fixture for Easy Retrievement in Case with Peri-Implantitis"

_materials, 2019, doi:10.3390/ma12020235_

Reviewer 1 Report

Dear authors,

Your paper entitled "Biomechanical study of separable implant for easy retrievement in case with peri-implantitis" explains why using separable implants in stage 2, in order  for easy retrivement in case of peri-implantitis, is better compared to using conventional ones. After carefully reading it, I have some comments and suggestions.

Please check if the title is correctly formulated in english language. Maybe with should be changed in of.

As one may understand, reading rows 31-36, your first choice in treating elderly patients is an implant. There is much to be discussed concerning this matter. Please rephrase this part of the introduction or give further explanations.

row 32. Also, several previously reported articles stated that missing teeth are closely related to death [2-4]. This is quite a hazardous statement. I understand that this is based on literature data, but certain conditions must be met in order to throw this conclusion. In my opinion, a reformulation is in order, or further explanation is needed.

row 108. A preload of 200N was applied vertically to the abutment screw to assume the condition of the fixation with the abutment and fixture [23]. Then, considering the load generated at various angles in the oral cavity, vertically 100N [25], 100N at 15 degrees [26], 30N at 45 degrees [25] of load were applied at the first molar (Fig 3). Please explain the choice of load value. Maybe not everyone would be interested in further reading the references.

Some punctual observations:

row 4. Eun sung Song. I presume capital is needed.

row 31. Dental defects and reductions primarily occur in the elderly population. Reduction in number of teeth causes poor eating habit and may deteriorate quality of life [1]. Please check if reduction is the most suitable therm, in this context. Maybe loss would be better.

row 99. Table 1. Material Properties of finite element models. Please replace Properties with properties.

There is no mention of Figure 1e in the text. Please add this.

row 116. FE analysis. Please use FEA, which was previously explained or explain FE.

Table 3,4,5. 15 deg. Is this a correct sign for degree? Please check and replace if necessary.

Peak von Mises stress. Please check if  peak or Peak is correct, and use same each time.

Please check your manuscript according to Instructions for authors. Some observations are as follow:

row 22 PVMS Abbreviations should be defined in parentheses the first time they appear in the abstract, main text, and in figure or table captions and used consistently thereafter.

Please redo the numbering of your references accordingly to Instructions for authors. References: References must be numbered in order of appearance in the text (including table captions and figure legends) and listed individually at the end of the manuscript. In your text the numbering goes until15, followed by 17-18, 29-31, goes back to 16, followed by 30-33, goes back to 19 etc.

row 72 etc. (Fig 1a,b) Please use Figure instead of Fig, Table instead of table, whenever necessary, according to Instructions for authors.

Author Response

Point1: Please check if the title is correctly formulated in english language. Maybe with should be changed in of. 

Response1:

We have discussed English language of this title, and changed the title as follow.

- row 2-3 : Finite element analysis of novel separable fixture for easy retrievement in case with peri-implantitis

 Point2: As one may understand, reading rows 31-36, your first choice in treating elderly patients is an implant. There is much to be discussed concerning this matter. Please rephrase this part of the introduction or give further explanations.

Response2:

Thank you for pointing out the sentence in rows 31-36.

We have deleted the sentence as we decided it was inappropriate to be in the introduction of this study. For a smooth context, we have rewritten the sentence as below.

- row 35-36 : Over the past several decades, treatment with dental implants has been widely used to restore the oral masticatory movement when natural teeth are lost or partially damaged [1-3].

 Point3: row 32. Also, several previously reported articles stated that missing teeth are closely related to death [2-4]. This is quite a hazardous statement. I understand that this is based on literature data, but certain conditions must be met in order to throw this conclusion. In my opinion, a reformulation is in order, or further explanation is needed.

Response3:  

We have removed this sentence as it was considered irrelevant to this study.

 Point4: row 108. A preload of 200N was applied vertically to the abutment screw to assume the condition of the fixation with the abutment and fixture [23]. Then, considering the load generated at various angles in the oral cavity, vertically 100N [25], 100N at 15 degrees [26], 30N at 45 degrees [25] of load were applied at the first molar (Fig 3). Please explain the choice of load value. Maybe not everyone would be interested in further reading the references.

Response4:  

Thank you for pointing out the sentence in row 108.

As you have mentioned, we write the reason of the selection for loading conditions in this study.

The load values used in this study were taken into account within the physiological load range. Then, we modified this sentence as follows.

- row 136-141 : A preload of 200N was applied vertically to the abutment screw to assume the condition of the fixation with the abutment and fixture [27]. Then, considering the load generated at various angles in the oral cavity, vertically 100N [28], 100N at 15° [29], 30N at 45° [28] of load were applied at the first molar (Figure 3). The applied load was within physiological range [36], and various loading directions were applied as non-axial loading affects major remodeling at the interface between the bone and implant [37].

 Point5: Some punctual observations: row 4. Eun sung Song. I presume capital is needed.

Response5:  

We have rewritten the author’s name on your suggestion.

 Point6: row 31. Dental defects and reductions primarily occur in the elderly population. Reduction in number of teeth causes poor eating habit and may deteriorate quality of life [1]. Please check if reduction is the most suitable therm, in this context. Maybe loss would be better.

Response6:  

We agree that the phrase is not suitable in this context, so we have removed it from the introduction.

Point7: row 99. Table 1. Material Properties of finite element models. Please replace Properties with properties.

Response7:  

We have rewritten the word on your suggestion.

 Point8: There is no mention of Figure 1e in the text. Please add this.

Response8:  

We have added explanation of Figure 1 (e) on your suggestion.

- row 94-96 : Figure 1. 3D CAD model of dental implant system. (a) Conventional fixture type, (b) Separable fixture type, (c) Abutment screw, (d) Abutment and (e) Assembled model of implant system (Left : exterior design of the fixture / Right : internal structure of the fixture).

 Point9: row 116. FE analysis. Please use FEA, which was previously explained or explain FE

Response9: 

We thank the reviewer for this insight.

We have revised finite element analysis as FEA in the abstract and unified it to FEA from then.

 Point10: Table 3,4,5. 15 deg. Is this a correct sign for degree? Please check and replace if necessary.

Response10: 

As your opinion, we have changed all ‘deg’ signs to ‘°.’

 Point11: Peak von Mises stress. Please check if peak or Peak is correct, and use same each time.

Please check your manuscript according to Instructions for authors. Some observations are as follow:

row 22 PVMS Abbreviations should be defined in parentheses the first time they appear in the abstract, main text, and in figure or table captions and used consistently thereafter.

Response11: 

Thank you for your good point.

We have mentioned Peak von Mises stress (PVMS) in the abstract and revised it as the abbreviation PVMS from then.

 Point12: Please redo the numbering of your references accordingly to Instructions for authors. References:References must be numbered in order of appearance in the text (including table captions and figure legends) and listed individually at the end of the manuscript. In your text the numbering goes until15, followed by 17-18, 29-31, goes back to 16, followed by 30-33, goes back to 19 etc.

Response12:   

Thank you for pointing out the order of reference due to our mischecking.

We have carefully revised and reordered the reference numbering.

 Point13: row 72 etc. (Fig 1a,b) Please use Figure instead of Fig, Table instead of table, whenever necessary, according to Instructions for authors.

Response13:   

As your opinion, we have changed all ‘Fig’ to ‘Figure.’

Reviewer 2 Report

lines 32-33. Off topic. Delete. 

line 35- Add more recent references (I would suggest these https://onlinelibrary.wiley.com/doi/abs/10.1111/cid.12666) 

lines 44-45 also surface of titanium with antimicrobial properties ( https://www.sciencedirect.com/science/article/pii/S1286011517302916) 

Discussion need to be improved and also I need clarification of the comcept of separable implant.

Author Response

Point1: lines 32-33. Off topic. Delete.

Response1: We agree and removed the sentence as it was considered irrelevant to this study.

 Point2: line 35- Add more recent references (I would suggest these https://onlinelibrary.wiley.com/doi/abs/10.1111/cid.12666)

Response2:

We have added more recent reference on your suggestion.

- row 35-36 : Over the past several decades, treatment with dental implants has been widely used to restore the oral masticatory movement when natural teeth are lost or partially damaged [1-3].

 Point3: lines 44-45 also surface of titanium with antimicrobial properties

(https://www.sciencedirect.com/science/article/pii/S1286011517302916)

Response3: 

We have added more recent reference on your suggestion.

- row 44-46 : To prevent and treat peri-implantitis, various studies are currently being progressed in the dental field such as local or systemic antibiotics injection, surgical treatments, and implant surface of titanium with antimicrobial properties [15-18].

 Point4: Discussion need to be improved and also I need clarification of the concept of separable implant.

Response4:  

Thank you for your suggestion.

We have added brief explanation on the concept of separable implant as below.

- row 199-206 : When peri-implantitis occurs after dental implant placement, treatments such as antibiotics injection and metallic curettes were used to remove contamination on the surface of the conventional fixture; but it is difficult to clear-up perfectly [19-20]. For this reason, this study designed a fixture that is separable in top and bottom parts so that in case of peri-implantitis, the contaminated top part could be removed and replaced with a new one. As only the contaminated part is replaced, it is surgically more convenient compared to the conventional fixture. Also, it has a higher success rate of clearing up compared to the conventional method, hence it is considered to provide a higher success rate of peri-implantitis treatment.

Reviewer 3 Report

Thanks for submitting this manuscript, which presents an interesting numerical study on dental implants development towards peri-implantitis prevention.

In general, I understand what the authors have done, I agree with the methods and I reckon that the discussion/conclusion are aligned with the results. It is interesting to see that the separable implant can be more stable than the conventional one, but we can't have a truly solid conclusion with only one new model. 

Before going into further appreciations, the level of Enligsh is poor at some points, and must be improved prior to publication. 

The first sentence of the abstract is not clear enough, "The purpose of this study was to evaluate the structural and biomechanical properties between the newly designed separable implant and the conventional implant for easy retrievement in case of peri-implantitis", which is not a good start. Authors must give some context and clarify the research question.

In the introduction, sentences in lines 38 or 45, for example, must be re-written:

"However, the high success rate of dental implants are still encountering problem" - maybe you want to say that even with a decent success rate, problems still arise?

"... many attempts have been made to derive appropriate treatment method" - research has been trying to find more effective treatments?

Still in the introduction, besides the abovementioned English language issues, there is no reference to the methods (finite element analysis) nor a real literature review on dental implant modelling. This must be improved.

In the Materials and Methods section, the most serious issue is the lack of information on the number of cases evaluated. There's just an insufficient indication that "CBCT images of normal Korean male adult skulls" (line 84). This make me wonder why exactly are the authors presenting a statistical study. Also, no convergence study is presented, nor a clear indication of the source for each material property used in the models (table 1). Details such as "tie contact" or "general contact" could be explained for a broader audience.

In the Results, table 5 and figure 6 could be improved: the first is not effective in sharing information with the readers, it could be enhanced by having the average differences between the 2 groups, while the second is very dense and hard to read, so it could be simplified (e.g., there is no evident gain from presenting the two bony layers).

Finally, I don't have much to comment on the Discussion and Conclusion, but it is my opinion that the end of the Discussion (from line 230) could be moved to the introduction, as it is more of a revision of the literature.

Author Response

Point1: In general, I understand what the authors have done, I agree with the methods and I reckon that the discussion/conclusion are aligned with the results. It is interesting to see that the separable implant can be more stable than the conventional one, but we can't have a truly solid conclusion with only one new model. 

Response1:

Thank you for your good point.

If equivalence between the two models was proved with the most fragile model among various sizes, it would show an identical tendency with other sizes. Therefore, this study has selected an implant model of the smallest in diameter and length, considering the worst case.

 Point2: Before going into further appreciations, the level of English is poor at some points, and must be improved prior to publication.

Response2:

We agree and carefully revised the level of English in this paper.

 Point3: The first sentence of the abstract is not clear enough, "The purpose of this study was to evaluate the structural and biomechanical properties between the newly designed separable implant and the conventional implant for easy retrievement in case of peri-implantitis", which is not a good start. Authors must give some context and clarify the research question.

Response3: 

Thank you for pointing out the first sentence in the abstract.

We have discussed and revised the sentence so that it could explain the background and purpose of this study.

- row 17-21 : Peri-implantitis is a common complication following dental implant placement, which may lead to bone loss and fixation failure. With the conventional fixture, it is difficult to perfectly clear-up the infection. To solve this, we have designed a separable fixture of which the top part is replaceable. This study aimed to compare the structural and biomechanical stability of the separable and conventional fixture.

 Point4: In the introduction, sentences in lines 38 or 45, for example, must be re-written:

"However, the high success rate of dental implants are still encountering problem" - maybe you want to say that even with a decent success rate, problems still arise?

"... many attempts have been made to derive appropriate treatment method" - research has been trying to find more effective treatments?

Response4:  

As you have mentioned, we agree that the sentence is not clear and rewritten it as below.

- "However, the high success rate of dental implants are still encountering problem"

row 39-40 : Majority of dental implants have a high success rate in long term [8-10]. Nonetheless, problems such as postoperative complications, union, and fixation failures occur.

- "... many attempts have been made to derive appropriate treatment method"

 This phrase was removed as it was considered inappropriate.

 Point5: Still in the introduction, besides the abovementioned English language issues, there is no reference to the methods (finite element analysis) nor a real literature review on dental implant modelling. This must be improved.

Response5:  

As you have suggested, we added reference on finite element analysis (FEA) and dental implant modeling as below.

- row 73-79 : As the separable fixture consists of top and bottom parts compared to the conventional fixture, it may have a disadvantage of low stability due to the small gap at the junction between the top-bottom parts when attached to the abutment, crown, and abutment screw. Yet, there is no reference in comparison to the conventional fixture since no cases have been reported so far with such fixture. Therefore, analysis to compare the structural stability between the conventional and separable fixture is required. Previous studies have performed comparative biomechanical evaluation using FEA to analyze by various changes such as in design and load [25-31]. This study was conducted to compare the biomechanical stability between the easily separable fixture in case of peri-implantitis and the commonly used conventional fixture using FEA.

 Point6: In the Materials and Methods section, the most serious issue is the lack of information on the number of cases evaluated. There's just an insufficient indication that "CBCT images of normal Korean male adult skulls" (line 84). This make me wonder why exactly are the authors presenting a statistical study. Also, no convergence study is presented, nor a clear indication of the source for each material property used in the models (table 1). Details such as "tie contact" or "general contact" could be explained for a broader audience.

Response6:  

We have revised the phrase in line 84 on your suggestion and found it was miswritten. In this study, one CBCT image of the investigator was used to implement one bone mode. For statistical analysis, the stress value of each node at the contact interface between the bone and implant was derived and compared.

- row 103-104 : a CBCT image of normal Korean male adult skull was cut into 0.25 mm thickness to obtain 2-dimentional images,

- row 152 : The Max EQS values of the cortical and cancellous bone of each node at the contact surface between the bone and fixture were presented as mean ± standard error of the mean (SEM), and two independent sample t-test was performed using the means for comparison between conventional and separable fixtures.

In terms of convergence study, a previous study which performed FEA on short dental implants reported that the ideal mesh size was 0.3mm; so, this study used a smaller size of 0.15 mm.

- row 119 : A study by Joao Paulo performed FEA with a short dental implant and derived an optimum mesh size of 0.3 mm through mesh convergence [35]. Accordingly, the maximum mesh size of the implant (abutment, fixture, abutment screw) in this study was set at 0.15 mm which is lower than the previous study, the contact interface between the bone and the implant was 0.15 mm. The number of elements, nodes and mesh sizes used in this study were shown in Table 2.

As you have suggested, we added reference on each material properties.

We added more detailed description on tie contact and general contact.

- row 127 : The interface of the cortical and cancellous bone, bones and fixture, and abutment and crown were applied “Tie contact”. The “Tie contact” type simulated perfect osseointegration in which the implant and the surrounding bone were fully integrated so that neither sliding nor separation in the implant-bone interface is possible [27]. The interface between the fixture, abutment, and abutment screw were applied “Frictional contact”. A friction coefficient of 0.5 was used [27]. Frictional contact implies that a gap between the implant parts might exist under an occlusal force [27].

 Point7: In the Results, table 5 and figure 6 could be improved: the first is not effective in sharing information with the readers, it could be enhanced by having the average differences between the 2 groups, while the second is very dense and hard to read, so it could be simplified (e.g., there is no evident gain from presenting the two bony layers).

Response7: 

As you have suggested, we revised and rearranged Figure 5 for better understanding.

This study used one CBCT image for the model to compare the PVMS values, so there was no average value.

 Point8: Finally, I don't have much to comment on the Discussion and Conclusion, but it is my opinion that the end of the Discussion (from line 230) could be moved to the introduction, as it is more of a revision of the literature.

Response8: 

As you have suggested, we rearranged the sentence from Discussion to Introduction.

- row 64 : In a study on marginal bone loss of 131 implant prostheses by Cappiello et al., 1.67 ± 0.37 mm marginal bone loss occurred for platform matching implants and 0.95 ± 0.32 mm for platform switching implants [24]. Likewise, marginal bone loss after implant is inevitable, and aesthetic problems may arise if it occurred on the anteriors.

Round  2

Reviewer 2 Report

Dear Authors, 

congratulations for performing the correction so soon and to comply in an excelent way to the suggestions. 

Reviewer 3 Report

Thanks for the revision efforts.